# Fabrication of Hierarchically Porous CuBTC@PA-PEI Composite for High-Efficiency Elimination of Cyanogen Chloride

**DOI:** 10.3390/molecules28062440

**Published:** 2023-03-07

**Authors:** Xuanlin Yang, Liang Lan, Chao Zheng, Kai Kang, Hua Song, Shuyuan Zhou, Shupei Bai

**Affiliations:** State Key Laboratory of NBC Protection for Civilian, Beijing 100191, China

**Keywords:** cyanogen chloride, metal-organic frameworks, CuBTC/polyacrylate composite, surface modification, stepwise impregnation layer-by-layer growth

## Abstract

Cyanogen chloride (CNCl) is highly toxic and volatile, and it is difficult to effectively remove via porous substances such as activated carbon due to the weak interaction between CNCl and the adsorbent surface. Developing a highly effective elimination material against CNCl is of great importance in military chemical protection. In this work, a new metal-organic framework (MOF) CuBTC@PA-PEI (polyacrylate-polyethyleneimine) composite was prepared and exhibited excellent CNCl elimination performance in the breakthrough tests. PEI was used for the functionalization of PA with amino groups, which is beneficial to anchor with metal ions of MOF. Afterward, the growth of MOF occurred on the surface and in the pores of the matrix by molecular self-assembly via our newly proposed stepwise impregnation layer-by-layer growth method. Breakthrough tests were performed to evaluate the elimination performance of the composites against CNCl. Compared with the pristine CuBTC powder, the CuBTC@PA-PEI composite exhibited better adsorption capacity and a longer breakthrough time. By compounding with the PA matrix, a hierarchically porous structure of CuBTC@PA-PEI composite was constructed, which provides a solution to the mass transfer problem of pure microporous MOF materials. It also solves the problems of MOF molding and lays a foundation for the practical application of MOF.

## 1. Introduction

Cyanogen chloride (CNCl) is one of the most difficult toxic gases to remove via adsorption because of its weak interaction with the adsorbents [1]. CNCl is one of the systemic toxic agents, having the closest melting and boiling temperatures [2]. It is fast-killing and lethal, which can strongly inhibit cell respiratory chain functions, causing obstacles to the use of oxygen and generating energy reduction. A low concentration of CNCl can cause severe symptoms such as tearing, sneezing, coughing, and vomiting [3]. The molecular size of cyanogen chloride is small (about 5.6 Å), and the pure physical adsorption capacity of porous substances such as activated carbon is poor [4]. Additionally, because of its acid-forming capability, it is representative of a group of volatile acid gases, and it can be removed by chemical reactions with the active sites on the adsorbent. Although the “Convention on the Prohibition of the Development, Production, Stockpiling and Use of Chemical Weapons and on Their Destruction” officially entered into force in 1997, it is still possible that CNCl could be used in terrorist attacks because of its wide range of sources and easy availability. Research regarding a more efficient and reactive sorbent for CNCl can lead to smaller filtration devices [5], which will be beneficial to future military chemical defense equipment. Therefore, it is of great significance to study efficient CNCl elimination materials for national defense, environmental protection, and other fields.

The elimination material is the core of the treatment of toxic and harmful gases. Progress in the research aimed at the development of materials affording CNCl elimination can be divided into two categories: carbon-based materials and non-carbon-based materials [6]. Carbon-based materials used in the field of CNCl elimination are impregnated activated carbon, i.e., activated carbon loaded with various active components [7,8,9]. Characterized by a huge specific surface area, activated carbon is currently the most widely used adsorbent. However, the functional groups of the impregnated carbon are difficult to modify according to researchers’ needs, and the pore structure of the activated carbon cannot be designed and regulated. In recent years, researchers have actively explored non-carbon-based elimination materials such as molecular sieves [10], zirconium hydroxide [5,11,12,13], and supported alumina catalysts [14,15,16]. In laboratory experiments, the performance of these materials is excellent, but most of them are in powder form, and the application of these materials is limited. Over the past decades, metal-organic frameworks (MOFs) have been widely studied and applied in the field of gas adsorption [17], separation [17], and catalysis [18]. MOFs have the advantages of high specific surface area, high porosity, and adjustable pore size and have broad application prospects [19]. MOF appears in powder form. If directly loaded into a fixed bed, there will be huge resistance to the gas flow. Additionally, MOFs are more difficult to process compared to traditional adsorbent materials, which hinders the industrial production and application of MOFs. In addition, the pure microporous structure of most MOFs is not conducive to the mass transfer of guest molecules, which restricts the application of MOFs in adsorption, catalysis, and other fields [20].

New design strategies and applications of MOF composite materials have been raised in recent years [21,22]. Researchers have proposed that by combining MOFs with another functional matrix, the advantages of the two materials could be coupled, and the inherent defects of MOFs could be overcome [23,24,25]. Among various substances, organic polymer has excellent mechanical properties, easy availability, controllable particle size and pore size, and low cost [26]. Polyacrylate (PA) beads prepared by the emulsion template method [27] have received extensive attention in the field of organic polymer porous materials, with a crosslinked interconnecting pore structure, easy permeability, high diffusivity, which is beneficial for mass transfer [28,29,30]. Characterized by good mass transfer properties, as well as flexible pore size and diverse functionalization strategy [31], PA beads are a promising matrix material. When combined with MOF, the resulting MOF/PA composite will not only solve the problem of excessive resistance when the powder MOF is loaded into the fixed bed but also enrich the pore types of the pure microporous MOFs, alleviating the problem of mass transfer hindered by the pure microporous structure of MOF [32] and thus reducing the bed resistance and improving the mass transfer ability of guest molecules in the practical applications.

Although the PA matrix has many excellent properties [27], the functional groups on the surface of the material only have epoxy groups and ester groups, which are not active enough for coordination with the metal ions of MOFs, resulting in the poor growth of MOFs. Therefore, it is necessary to functionalize the PA matrix. Polyethyleneimine (PEI) is a water-soluble polymer produced by the polymerization of ethylene imine. It is not a completely linear polymer but a partially branched polymer containing a primary amine, secondary amine, and tertiary amine. With strong reactivity due to a large number of polar groups (amino), PEI could be combined with different substances. After impregnation, PEI can strongly interact with the functional groups of the matrix materials. Additionally, the amino groups can be coordinated with most metal ions of MOFs, so PEI is a perfect modifier.

Most scholars only tested the breakthrough time of CNCl for a representative of the elimination capabilities of CNCl but did not study the entire breakthrough process [4,5,33,34]. For instance, Naderi et al. [35] studied new adsorbents based on MCM-41, which was modified with Cu^2+^ and triethylenediamine (TEDA) for CNCl elimination. They only tested the breakthrough time and made a comparison with the impregnated carbon. However, the breakthrough time of Cu^2+^-TEDA modified MCM-41 was only 4 min, which was much shorter than that of the impregnated carbon (29 min). Peterson et al. [33] synthesized UiO-66-NH_2_ in a scaled batch and then pressed the UiO-66-NH_2_ powder into small pellets to test the elimination performance of CNCl. However, the breakthrough times for cyanogen chloride were dramatically short at nearly zero, likely as a result of mass-transfer limitations from the completely microporous MOF.

As a result, scholars did not fully understand the law of the whole breakthrough process of CNCl, and they also did not make clear the reasons for the insufficient protective ability of various materials against CNCl, which is not conducive to the research of new materials to improve the elimination capacities towards CNCl.

In this study, a hierarchically porous structure CuBTC MOF/PA composite, abbreviated as CuBTC@PA-PEI, with highly loaded CuBTC was prepared by our newly proposed stepwise impregnation layer-by-layer (LBL) growth method. The polyacrylate spherical matrix with macropores was synthesized by emulsion polymerization. By surface modification, a large number of Lewis alkaline sites were distributed on the surface and in the pores of the spherical matrix. Afterward, the growth of CuBTC occurred by molecular self-assembly. Breakthrough tests were performed to evaluate the elimination capacities of the composites against cyanogen chloride. Our work will help solve the problems of molding and packaging MOFs and lay a foundation for the practical application of MOFs.

## 2. Results and Discussion

### 2.1. Structural Characterization

Figure 1 shows the morphology of the PA beads as well as the pore structures on the surface and in the sphere. The PA beads have a regular spherical shape (Figure 1a), which is conducive to filling the bed and reducing the void fraction of the bed. The diameter of most beads is between 0.4 mm and 0.8 mm. As shown in Figure 1b,c, the pores show macroporous characteristics and have interconnecting channels. The size of most pores is about 2~3 μm, and these macropores are interconnected on the pore walls. The interconnected macroporous structure of the PA matrix provides sufficient space for MOF to grow without causing a blockage.

SEM images of the CuBTC@PA-PEI samples prepared under different process conditions are shown in Figure 2. The loading rate and uniformity of MOF on composites prepared by different methods vary from each other. The one-pot method is simple and convenient. Due to the rapid CuBTC nucleation rate in the liquid phase, a large number of CuBTC grains quickly formed when the copper salt solution was mixed with the trimesic acid solution. Therefore, fewer grains grew on the matrix (Figure 2a,b), and the MOF loading rate of the composite was low. In the stepwise impregnation growth method, we first impregnated PA-PEI with an ethanol solution of copper nitrate and immersed it for 12 h so that the copper nitrate solution could fully be diffused into the pore and the amine groups on PA-PEI were coordinated with copper ions. As shown in Figure 2c,d, the number of MOF grains on the surface and in the pores of the composite is higher than that of the grains prepared by the one-pot method. As shown in Figure 2e,f, in our newly proposed stepwise impregnation LBL growth method, the CuBTC grains on the surface and inside the matrix increased obviously and grew uniformly. The distribution of elements on the cross-section of beads prepared by the stepwise impregnation LBL growth method was analyzed by EDS mapping. As can be seen in Figure 2g,h, the distribution of N elements and Cu elements on the cross-section of the bead was uniform, indicating that the impregnation liquid could fully diffuse into the beads. The distribution area of the N element and Cu element on the cross-section was almost the same, which indicated that the copper ions were combined with the amine groups.

Hygroscopicity is an important property of solid sorbents that is coherently related to storage stability. The hygroscopicity of the PA matrix and the composites was evaluated by measuring the water contact angle. As shown in Figure 3, the water contact angle of the PA matrix is obtuse (109.71°), revealing that the PA matrix is hydrophobic. After the functional modification of amine groups, water droplets spread on the surface of the material, and the water contact angle was reduced to 0°, indicating that the amino-functionalized material is hydrophilic. It can be seen that the amine groups have been successfully loaded onto the PA matrix. The functionalization of amine groups changed the surface property of the PA matrix and improved the compatibility of MOF and the PA matrix. After the growth of CuBTC, the water contact angle increased to 28.35°, and the hydrophobicity increased.

XRD measurements were conducted to determine the crystallinity of the composites. Figure 4a shows the XRD patterns of the PA, CuBTC@PA-PEI (1 layer), CuBTC@PA-PEI (2 layers), pristine CuBTC powder, and simulated CuBTC. The PA matrix is an amorphous structure, and the XRD spectrum shows a bulge peak. The patterns of the CuBTC@PA-PEI (1 layer) and the CuBTC@PA-PEI (2 layers) composites exhibited similar Bragg reflections to those of pristine CuBTC powder, which conform to the simulated pattern of the CuBTC. This confirms that the CuBTC structure remained intact. As vividly shown in Figure 4a, the intensity of the characteristic peaks in the CuBTC@PA-PEI (2 layers) was higher than that in the CuBTC@PA-PEI (1 layer). This reflects that the stepwise impregnation LBL growth method can greatly improve the crystallinity of MOF. As shown in Figure 4b, XRD patterns of CuBTC@PA-PEI before CNCl adsorption and after exposure to CNCl. The XRD patterns show the same peak shape, indicating that the structure of CuBTC did not change after adsorption. The thermal stability of the composite was tested by thermogravimetric analysis. As shown in Figure 4b, the CuBTC@PA-PEI compound can remain stable below 200 °C and is suitable for use at room temperature, even in high temperatures.

The FTIR spectra shown in Figure 5 provide clear evidence for the successful synthesis of CuBTC and the amino-modified PA matrix. For the PA matrix, the characteristic peaks at 849 cm^−1^ and 909 cm^−1^ are attributed to the epoxy group stretching vibration [32]. The peak at 1733 cm−1 is ascribed to the ester carbonyl group stretching vibration [32]. It can be seen that the main functional groups on the PA matrix are epoxy groups and ester groups. After surface modification, the peak at 909 cm^−1^, which represents the epoxy group, disappears. The peaks at 1574 cm^−1^ and 3420 cm^−1^ represent N-H bond deformation vibration, and the peaks are broad and strong [32]. Under the condition of 130 °C, the epoxy group on the PA matrix reacts with the amine group of the PEI, and the epoxy group opens the ring, so the characteristic peak of the epoxy group disappears. This also proves that the amine group is successfully loaded onto the PA matrix. After compounding with CuBTC, a peak at 730 cm^−1^, which represents the Cu-O bond stretching vibration, appeared. Additionally, the C=O stretching vibration shifted to 1643 cm^−1^ due to the coordination of carboxyl with Cu^2+^, and the characteristic vibration of C-C bonds in the benzene ring appeared at 1375 cm^−1^ [32]. This evidence indicated the successful generation of CuBTC in the PA matrix.

N_2_ adsorption-desorption experiments were performed to assess the porosity during the preparation process of the MOF@PA-PEI composites. The isotherms and DFT pore size distributions are shown in Figure 6. As shown in Figure 6a,b, macropores made up the vast majority. After PEI modification, the macropore ratio of PA matrix decreased, which may be due to the accumulation of macromolecular polymer PEI in the pore, blocking the macropore structure. After the generation of CuBTC, the isotherms of the composites are featured by typical Type IV sorption behavior, which is hierarchical porous materials with microporosity of MOF as well as interparticle slit-shaped mesoporosity. As shown in Table 1, the specific surface area of the PA matrix is only 52 m^2^·g^−1^, and it has a macropore structure, which can provide large enough space for the growth of MOF grains. After surface modification with PEI, some pores may be blocked by the large polymer molecule PEI, and the specific surface area decreased to 34 m^2^·g^−1^. However, with high-density amino groups on the PEI molecules to anchor metal ions, the loss of specific surface area is not of great significance. The BET surface area of CuBTC@PA-PEI (2 layers) was 471 m^2^·g^−1^. DFT pore size distribution data showed that the CuBTC@PA-PEI (2 layers) had a higher content of micropores than that which was loaded with one layer of CuBTC. After the composites were exposed to CNCl, the content of micropores decreased, and the BET surface area decreased from 471 m^2^·g^−1^ to 236 m^2^·g^−1^ as well, which probably represented the destruction of some pore structures after CNCl adsorption.

### 2.2. Elimination Abilities for CNCl

We tested the performance of the prepared composites toward CNCl, which is a highly toxic blood agent, and one of the most difficult CWAs to remove. Breakthrough time and adsorption capacities are important indexes to evaluate the elimination abilities of the prepared materials. Figure 7 presents the CNCl breakthrough curves of the composites prepared by different methods as well as the PA-PEI matrix and pristine CuBTC powder as a contrast. The results are calculated and summarized in Table 2. The PA-PEI matrix has no elimination property for CNCl. This may be attributed to the poor adsorption performance of the macroporous structure of PA-PEI, which results from the low surface area. When the CNCl flowed through the adsorbent bed of pristine CuBTC powder, it penetrated immediately, and the breakthrough time was zero. This may be ascribed to the damaged pore structure of MOF due to tableting, resulting in a serious decrease in adsorption properties. As shown in Table 2, the elimination abilities of materials are closely related to the load rate of MOF. The CuBTC@PA-PEI composite prepared by the stepwise impregnation LBL growth method, which is referred to as CuBTC@PA-PEI (2 layers), exhibited the longest breakthrough time among the composites prepared by other methods. The breakthrough time of the CuBTC@PA-PEI (2 layers) sample is about five times longer than that of the CuBTC@PA-PEI prepared by the one-pot method and double that of the CuBTC@PA-PEI prepared by the stepwise impregnation growth method.

Despite the elimination performance of the composite towards CNCl, the reversibility of CNCl by CuBTC@PA-PEI was evaluated. The composite was regenerated at 120 °C. As shown in Figure 8, the CuBTC@PA-PEI had an excellent elimination capacity for CNCl. As the number of regenerations increased, the breakthrough time decreased. The breakthrough time decreased by ~50% after the composite had been regenerated five times. Then, the XRD of the CuBTC@PA-PEI composite after CNCl breakthrough was taken to examine the stability of the composite (Figure 8b). The XRD result shows that after five cycles of regeneration, the characteristic peak intensity of CuBTC was greatly reduced, which reveals the destruction of the CuBTC structure. These results indicated that although the elimination performance of the CuBTC@PA-PEI composite is excellent, the stability is not so good. The structure of CuBTC was destroyed during the CNCl breakthrough process. This will be the focus of future work to study. Only by solving the stability problem can the practical application of this composite material become possible.

In military chemical protection, breakthrough time is much more significant than adsorption capacity. The breakthrough time of composites prepared in this work is compared with other materials reported in the literature. The results are summarized in Table 3. The breakthrough time of CuBTC@PA-PEI (2 layers) prepared in this work is longer than that of other materials. MOF/Polymer composites are promising to be used as CNCl elimination materials.

## 3. Experimental Section

### 3.1. Materials

Glycidyl methacrylate (GMA, >99%), Trimethylolpropane triacrylate (TMPTA, >99%), Tert butyl methacrylate (TBMA, >99%), Cetyl alcohol (>99%), Benzoyl peroxide (>98%), Toluene (>99%), N, N-Dimethylaniline (>99%), Ammonium persulfate (>99%), Polyvinyl alcohol (PVA, molecular weight 1788, >99%), ethanol (>99%), methanol (>99%), Polyethyleneimine (98%), Trimesic Acid (>99%), and Tetramethylethylenediamine (>99%) were purchased from Macklin (Shanghai, China). PEG-PPG-PEG (>99%) and Cu(NO_3_)_2_·3H_2_O (99%) were purchased from Sigma Aldrich Inc. (St. Louis, MO, USA). Nitrogen (99.999%) was purchased from Haipu Gas Co., Ltd. (Beijing, China). All the chemicals were not further treated before use.

### 3.2. Materials Preparation

The preparation process of MOF@PA-PEI composites can be divided into three steps. One is the synthesis of the PA matrix, another is the functional modification of the PA matrix, and the third is the growth of MOF on the modified matrix. The schematic diagram for the synthesis process of MOF@PA-PEI is shown in Figure 9. 

#### 3.2.1. Synthesis of PA

The preparation process of the PA matrix is mainly formed by two key steps: emulsion preparation and suspension polymerization [30]. The recipe for the synthesis of the PA matrix is shown in Table 4.

*1.* 
*The First Step: Emulsion Preparation*


Glycidyl methacrylate (GMA), tert butyl methacrylate (TBMA), and trimethylolpropane triacrylate (TMPTA) were dissolved in toluene according to the recipe shown in Table 4. The oil-soluble polyoxyethylene polyoxypropylene copolymer (PEG-PPG-PEG, referred to as P123) was used as an emulsifier, which was fully dissolved by ultrasound to form the oil phase. The water-in-oil (W/O) emulsion was prepared by dropping water into the oil phase with a high-speed disperser (3000 rpm for 3 min, then 6000 rpm for 2 min, and 8000 rpm for 5 min). The whole process was carried out in an ice water bath below 0 °C.

*2.* 
*The Second Step: Suspension Polymerization*


The third phase was prepared with 300 mL water, 2 g polyvinyl alcohol, and 0.48 g ammonium persulfate, which were placed in a three-necked flask in an oil bath at 70 °C. Then, the W/O emulsion prepared in the first step was added into the three-necked flask and stirred at 270 rpm foCr 10 min, during which nitrogen was introduced to prevent oxygen inhibition. The particle size could be controlled by adjustment of the stirring speed. Afterward, a small amount of reductant N, N-tetramethylethylenediamine was added to control the polymerization. Because of the interfacial tension, the emulsion formed spherical particles in the third phase. At this time, a large number of white beads appeared in the third phase. Then, the white beads were washed with water and ethanol three times, respectively, to remove the residual emulsifier in the pores of the beads. After these two steps, the spherical PA beads were prepared. The reaction process to synthesize PA is shown in Figure 10.

#### 3.2.2. Surface Modification

The macroporous PA beads were put into a round-bottomed flask and degassed at 120 °C for 2 h to facilitate the next operation, as the pressure difference makes it easier for the impregnation solution to spread to the internal channels. Then, a solution of PEI in methanol (mass ratio 4:1) was added into the flask and kept at 130 °C for 4 h. Finally, a Soxhlet extraction was carried out with methanol at 60 °C for 12 h and dried in a vacuum oven at 70 °C for 12 h. The PA matrix material modified by PEI was named PA-PEI. The reaction process of surface modification of the PA matrix is shown in Figure 11.

#### 3.2.3. Growth of CuBTC/PA Composites

We proposed three strategies for the growth of CuBTC/PA composites: the one-pot method, the stepwise impregnation growth method, and the stepwise impregnation LBL growth method. ① In the one-pot method, the metal salt solution (Cu(NO_3_)_2_·3H_2_O, 0.1 M, in ethanol) and the ligand solution (Trimesic acid, 0.05 M, in ethanol) were immersed into *PA-PEI* simultaneously. Then, the impregnated *PA-PEI* beads along with the CuBTC/PA precursor solution were loaded into a Teflon reactor for a solvothermal reaction at a temperature of 85 °C. ② The stepwise impregnation growth method means that the metal salt solution was first impregnated into the *PA-PEI* beads and dried to enrich the metal ions on the surface and in the pores of the beads. Then, the dried beads loaded with metal ions were immersed in the ligand solution for the self-assembly of the metal ion and the ligand on the surface and in the pores of *PA-PEI* beads. Then, the solvothermal reaction was carried out the same as the one-pot method. The sample prepared by this method was abbreviated as CuBTC*@PA-PEI* (1 layer). ③ The stepwise impregnation LBL growth method means that based on the as-synthesized *MOF@PA-PEI*, the process of impregnation and the solvothermal reaction were carried out again to enrich metal ions on the original *MOF*, and then the self-assembly with ligands occurred to form a second layer of *MOF*. The sample prepared by the stepwise impregnation LBL growth method was abbreviated as CuBTC*@PA-PEI* (2 layers). After the growth of *MOF*, the composite beads were immersed in methanol to remove excess ligands and then dried in a vacuum at 120 °C for 24 h. The load rate of *MOF* was calculated as the equation below:$$\mathrm{load}\;\mathrm{rate}=\frac{m_{MOF@PA-PEI}-m_{PA-PEI}}{m_{PA-PEI}}\times 100\%$$

#### 3.2.4. Synthesis of CuBTC Powder

CuBTC powder was synthesized according to the reported literature [36]. Trimesic Acid (0.21 g, 1 mmol) and Cu(NO_3_)_2_·3H_2_O (0.48 g, 2 mmol) were dissolved in 15 mL ethanol, respectively. In the next step, the two solutions were mixed into a Teflon reactor and placed in an oven at 85 °C for 24 h. After cooling, the sample was washed three times with ethanol and soaked for 48 h. Finally, the sample was dried at 120 °C for 24 h. The dried dark blue powder is CuBTC.

### 3.3. Characterization

Scanning electronic microscopy (SEM) images were acquired on a TESCAN Mira 4 scanning electron microscope equipped with Energy-dispersive X-ray spectroscopy (EDS). The observation was conducted at an acceleration of voltage of 1.5 kV, using samples whose fractured surfaces were coated with a thin gold layer before observation. Fourier-transform infrared spectroscopy (FTIR) was acquired on the Thermo Scientific Nicolet iS20 infrared spectrometer. X-ray diffraction (XRD) patterns were recorded with an X-ray diffraction system with Cu Kα radiation, working at 40 kV and 40 mA on the Thermofisher Nexsa X-ray diffractometer. The samples were degassed at 120 °C for 12 h and then used for N_2_ adsorption isotherm measurement at 77 K with a Quantachrome Autosorb-1-C instrument. The surface area, SBET (multipoint Brunauer—Emmett—Teller method), total pore volume, and pore size distribution (density functional theory, DFT) were acquired from the isotherms. Thermogravimetric analysis (TGA) was carried out by a thermogravimetric analyzer Rigaku TG/DTA8122. The test temperature was changed from 30 °C to 800 °C and at a heating and cooling rate of 10 °C/min under the nitrogen atmosphere.

### 3.4. Breakthrough Testing

Figure 12 shows the schematic diagram of the cyanogen chloride breakthrough time test device.

The sorption abilities of the composites towards CNCl were measured by breakthrough tests. Breakthrough tests were performed using bead samples on a self-built breakthrough apparatus. The breakthrough apparatus consisted of a gas generation module and an analysis module. Due to the corrosivity of CNCl, a capillary flowmeter was used to indicate the flow. The CNCl gas volatilized from the steel cylinder, flowing through the capillary flowmeter under the control of the valve, and was evenly mixed with air in the mixing chamber. Then, it passed through the adsorption bed at a certain flow rate, and the gas chromatograph with an FID detector was used to monitor the tail gas concentration.

The prepared sample was activated in a vacuum at 120 °C overnight. The pristine CuBTC powder was pressed into pellets, then crushed and sieved into 20 to 40 mesh particles, which were then loaded into quartz tubes for testing. The composite beads were loaded directly into a quartz tube and then compacted so that CNCl would not escape from the bed void. A stream from the CNCl cylinder was mixed with airflow to reach the challenge concentration (1600 ppm) and sent through the adsorbent bed. The breakthrough test parameters are shown in Table 5. The endpoint (breakthrough concentration) was 2 ppm. The adsorbed amount of CNCl was calculated using the adsorption time to the breakthrough concentration. The breakthrough capacity was calculated based on the integration of the area above the breakthrough curves.

## 4. Conclusions

In this study, the hierarchically porous CuBTC@PA-PEI was successfully prepared by our newly proposed stepwise impregnation LBL growth method. This new preparation strategy greatly improved the loading rate of MOFs on the organic polymer matrix and the PEI as a modifier promoted the combination of MOF and organic polymer matrix. According to XRD, FTIR, and TG analysis, CuBTC has been successfully anchored to the PA matrix, and maintains a complete crystalline structure, showing good thermal stability. In addition, we first reported that MOF/PA composites can work as a highly effective CNCl elimination material. In comparison with the pristine CuBTC powder, CuBTC@PA-PEI exhibited better adsorption performance of CNCl. This study helps solve the problems of molding and packaging MOF powder and lays a foundation for the practical application of MOF. We believe the results presented here offer new insight into the design of novel adsorbents for toxic chemicals, which are difficult to remove via conventional adsorbents.

## Figures and Tables

**Figure 1 molecules-28-02440-f001:**
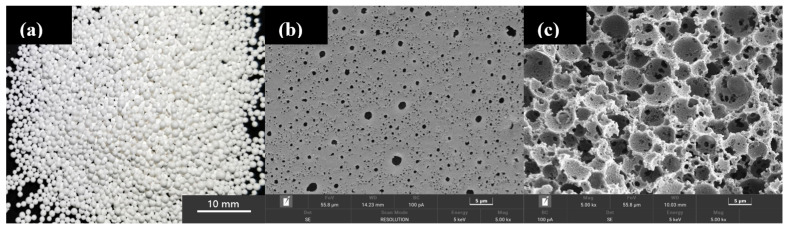
(**a**) Optical photograph of PA beads; (**b**) SEM image of the surface of PA beads; (**c**) SEM image of the cross-section of PA beads.

**Figure 2 molecules-28-02440-f002:**
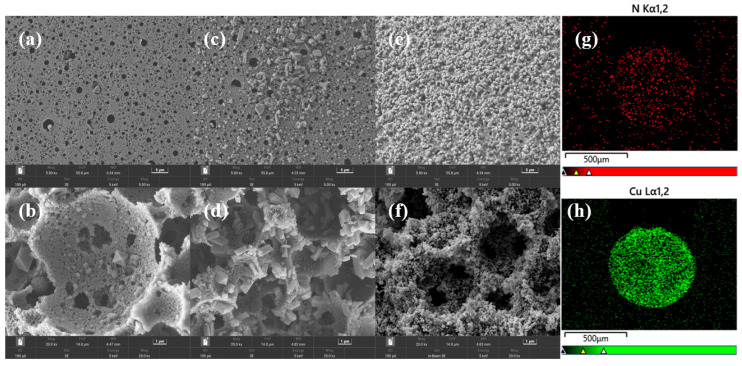
The surface of the beads prepared by (**a**) the one-pot method; (**c**) the stepwise impregnation growth method; (**e**) the stepwise impregnation LBL growth method. The internal pores of the beads prepared by (**b**) the one-pot method; (**d**) the stepwise impregnation growth method; (**f**) the stepwise impregnation LBL growth method. EDS mapping images of the cross-section of CuBTC@PA-PEI beads prepared by stepwise impregnation LBL growth method (**g**) N element; (**h**) Cu element.

**Figure 3 molecules-28-02440-f003:**
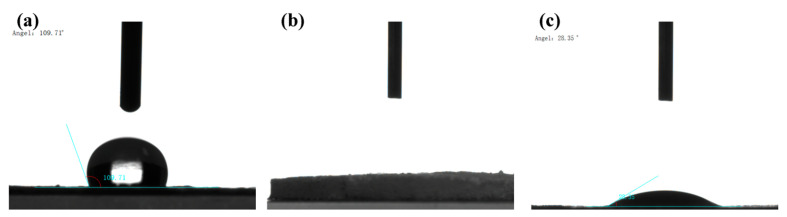
Contact angle with water of (**a**) PA; (**b**) PA-PEI; (**c**) CuBTC@PA-PEI.

**Figure 4 molecules-28-02440-f004:**
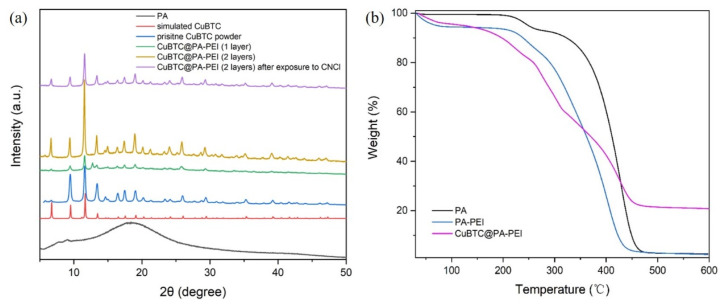
(**a**) The XRD patterns of PA matrix, CuBTC@PA-PEI (1 layer), CuBTC@PA-PEI (2 layers), simulated CuBTC, and CuBTC@PA-PEI after exposure to CNCl; (**b**) TGA curves of PA matrix, PA-PEI, and CuBTC@PA-PEI.

**Figure 5 molecules-28-02440-f005:**
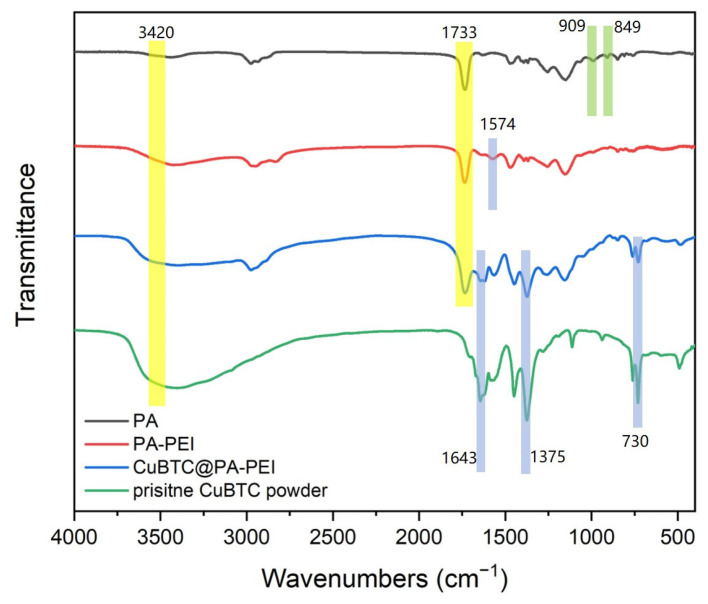
FTIR spectra of PA matrix, PA-PEI, CuBTC@PA-PEI, pristine CuBTC powder.

**Figure 6 molecules-28-02440-f006:**
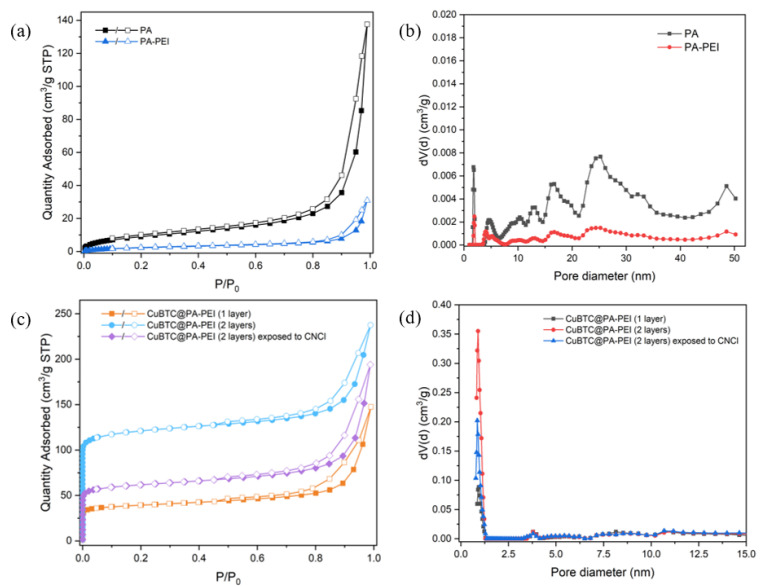
(**a**) N_2_ adsorption-desorption isotherms of PA and PA-PEI; (**b**) DFT pore size distributions of PA and PA-PEI; (**c**) N_2_ adsorption-desorption isotherms of CuBTC@PA-PEI (1 layer), CuBTC@PA-PEI (2 layers) and CuBTC@PA-PEI (2 layers) exposed to CNCl; (**d**) DFT pore size distributions of CuBTC@PA-PEI (1 layer), CuBTC@PA-PEI (2 layers) and CuBTC@PA-PEI (2 layers) exposed to CNCl.

**Figure 7 molecules-28-02440-f007:**
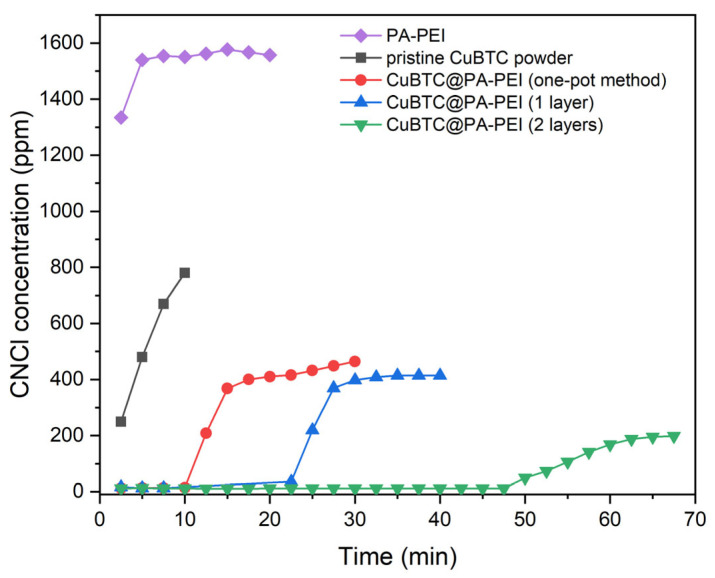
Breakthrough curves of CuBTC@PA-PEI and pristine CuBTC.

**Figure 8 molecules-28-02440-f008:**
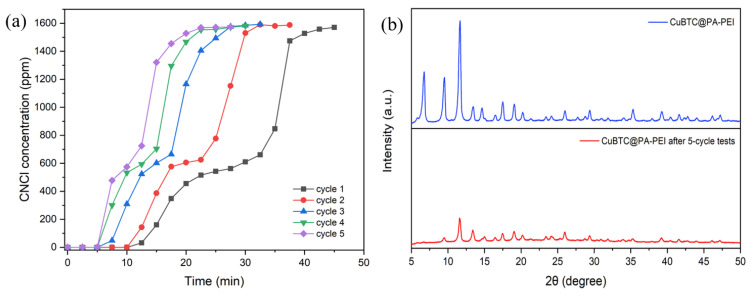
(**a**) CNCl breakthrough curves of CuBTC@PA-PEI for different adsorption-desorption cycle times; (**b**) XRD patterns of CuBTC@PA-PEI before and after 5 cycles of adsorption-desorption.

**Figure 9 molecules-28-02440-f009:**
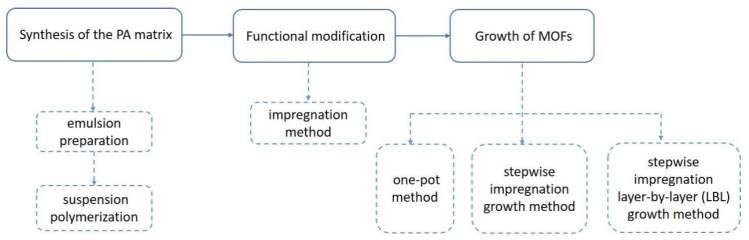
The schematic diagram for the synthesis process of MOF@PA-PEI.

**Figure 10 molecules-28-02440-f010:**
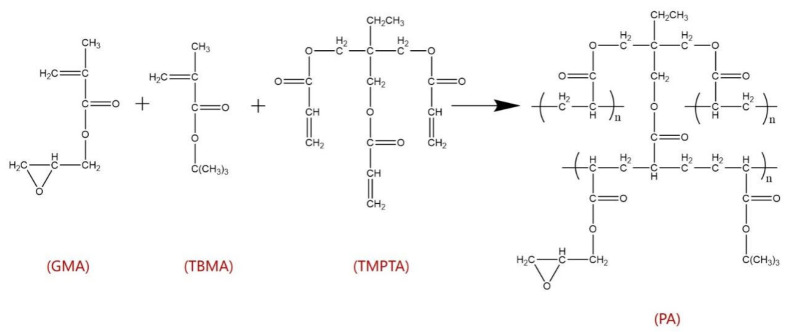
The reaction process to synthesize PA.

**Figure 11 molecules-28-02440-f011:**
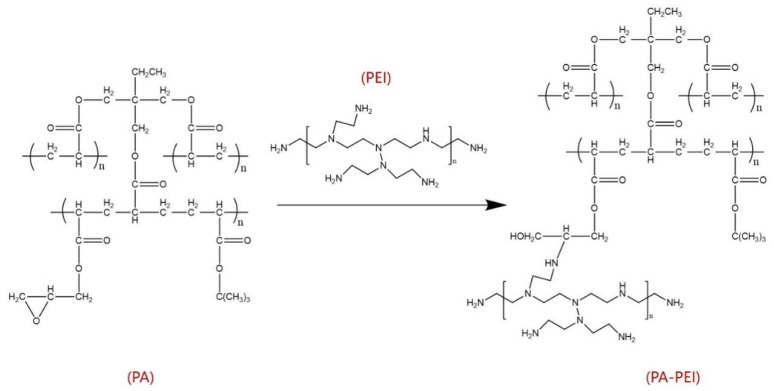
The reaction process of surface modification of PA.

**Figure 12 molecules-28-02440-f012:**
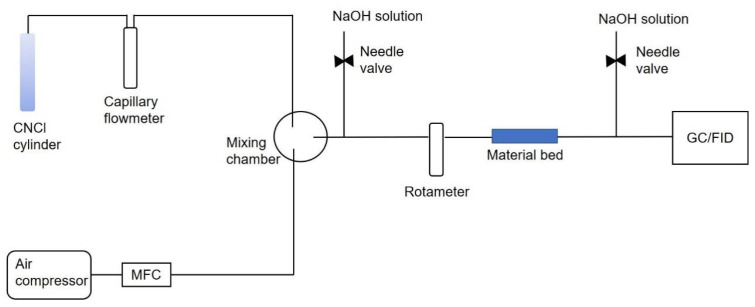
Schematic diagram of the device for cyanogen chloride breakthrough tests.

**Table 1 molecules-28-02440-t001:** S_BET_ of the PA, PA-PEI, and CuBTC@PA-PEI samples before and after the CNCl exposure.

Sample	S_BET_ (m^2^/g)
PA	52
PA-PEI	34
CuBTC@PA-PEI-1 layer	149
CuBTC@PA-PEI-2 layers (Before)	471
CuBTC@PA-PEI-2 layers (After)	236

**Table 2 molecules-28-02440-t002:** Load rate and elimination abilities of materials.

Sample	Load Rate (%)	Breakthrough Time (min)	CNCl Capacity (mmol·g^−1^)
CuBTC@PA-PEI (one-pot method)	16	10	0.08
CuBTC@PA-PEI (1 layer)	20	22	0.18
CuBTC@PA-PEI (2 layers)	52	47	0.39

**Table 3 molecules-28-02440-t003:** Breakthrough time comparison of other materials reported in the literature.

Material Type	Sample	Testing Condition	Breakthrough Time (min)	Ref.
Molecular sieve	Cu^2+^-SiAlMCM-41-TEDA	Challenge concentration: 3200 ppm. Others not mentioned.	4	[35]
MOF	UiO–66–NH_2_–5K	Challenge concentration: 1600 ppm. Breakthrough concentration: 2 ppm. Others not mentioned.	0	[33]
	MOF-808		0	[4]
Activated carbon	Carbon–Cu–Cr–TEDA		33	[5]
	10% TEDA/carbon		0	[5]
Metal hydroxide	Zr(OH)_4_		0	[5]
MOF composite	CuBTC@PA-PEI (2 layers)		47	This work

**Table 4 molecules-28-02440-t004:** The synthesis recipe of porous PA.

Reagent	GMA	TBMA	TMPTA	Toluene	P123	BPO	Cetyl Alcohol	H_2_O
Mass percentage (wt%)	9.3	2.8	8.1	15.6	2.4	0.4	0.4	61

**Table 5 molecules-28-02440-t005:** Cyanogen chloride breakthrough test parameters.

Breakthrough Parameter	Value
Challenge concentration	1600 ppm
Temperature	20 °C
Bed weight	0.7 g
Flow rate	100 mL/min

## Data Availability

Not applicable.

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
