# Peer review of "Fabrication of Hierarchically Porous CuBTC@PA-PEI Composite for High-Efficiency Elimination of Cyanogen Chloride"

_molecules, 2023, doi:10.3390/molecules28062440_

Round 1

Reviewer 1 Report

The manuscript entitled “Fabrication of hierarchically porous CuBTC@PA-PEI composite for high-efficient elimination of cyanogen chloride” is an interesting account of metal-organic framework viz CuBTC and PEI composite formation and their application in toxic gas capture. Although the concept is interesting, the manuscript lacks proper material characterization and drawing conclusions thereof. Moreover, experimental results do not fully contribute to supporting the data and claims made in the manuscript. I recommend a major revision of the manuscript complying with the following points.

1) The main concern is the stability of the material under testing. CuBTC has bad stability in bases, and PEI might be responsible for the partial degradation of the framework as the parent MOF peak appears very low intensity compared to literature one.

2) Continuing from previous doubt, the surface area reduction is not clearly evident to arise from which factor, whether from the degradation of MOF pores or polymer incorporation, or both. The pore size distribution is also not clearly presented to look at the actual state of the porosity of the materials after PEI incorporation.

3) Finally, the recyclability test is missing which poses a serious threat to the final state of the materials. Not only performance tests but also final material stability tests needed to be performed in order to know the real utility of the samples.

Reviewer 2 Report

In this paper, Yang et al. proposed and synthesized new MOF/polymer composite, which exhibited excellent cyanogen chloride elimination performance in the breakthrough tests. Overall, this work has performed very well in detailed experimental studies and has important implications for the study of MOFs in the context of hazardous gas adsorption. I would like to recommend it for publication in Molecules after the following point can be well addressed.

1. The authors should supplement the PA-PEI matrix as a control experiment for the cyanogen chloride breakthrough test.

2. PA-PEI (polyacrylate-polyvinylidene) is incorrectly named in the abstract.

3. The full names of FTIR and EDS are incorrect: FTIR stands for Fourier-transform infrared spectroscopy, and EDS is Energy-dispersive X-ray spectroscopy.  

4. The scale bar should be clear indicated in both optical and SEM images

4. Reference selection is good in the manuscript. It is recommended to add recent articles on design strategies and applications of MOF materials. (e.g., 10.1021/jacs.1c11750, 10.1002/anie.202105830).

Round 2

Reviewer 1 Report

The comments mentioned in the manuscript have been addressed to a satisfactory level. The manuscript may be published in my opinion.